systems biology

ageing, menopause, population genetics, kin selection

**Author for correspondence:**
Tin Yau Pang
e-mail: pang@hhu.de

# On age-specific selection and extensive lifespan beyond menopause

## Tin Yau Pang

Computational Cell Biology Group, Heinrich Heine University Düsseldorf, 25.02 02.33, Universitätsstr 1, Düsseldorf 40225, Germany

TYP, 0000-0003-4738-4032

Standard evolutionary theory of ageing predicts weaker purifying selection on genes critical to later life stages. Prolonged post-reproductive lifespan (PPRLS), observed only in a few species like humans, is likely a result of disparate relaxation of purifying selection on survival and reproduction in late life stages. While the exact origin of PPRLS is under debate, many researchers agree on hypotheses like mother-care and grandmother-care, which ascribe PPRLS to investment into future generations—provision to one's descendants to enhance their overall reproductive success. Here, we simulate an agent-based model, which properly accounts for age-specific selection, to examine how different investment strategies affect the strength of purifying selection on survival and reproduction. We observed in the simulations that investment strategies that allow a female individual to remain contributive to its own descendants (infants and adults) at late life stages may lead to differential relaxation of selection on survival and reproduction, and incur the adaptive evolution of PPRLS.

## 1. Introduction

The theory of evolution posits that natural selection favours the traits that enhance one's reproductive success, which depends not only on how frequently one reproduces but also on time and effort invested into the future generations [1]. The evolutionary theory of ageing further predicts differential selection at different stages of life. Because old individuals have lower overall reproductive output and also lower survival prospect than young ones, the strength of selection on age-specific loci weakens with increasing age. Therefore, deleterious mutations affecting early life stages tend to be removed by purifying selection, whereas those affecting late life stages tend to accumulate [2–6]

In most animal species, the end-of-reproduction coincides with the end-of-survival. There are a few exceptions, such as humans and resident killer whales, whose female individuals experience prolonged post-reproductive lifespan (PPRLS), i.e. a

significant portion of the female individuals in the population survive a substantially long period of time after their last birth [4,5,7,8]. Many researchers agree that investment in unweaned future generations is the primary cause of PPRLS, at least for humans. The qualitative explanation is like this: (i) human babies are born with head size close to the limit of safe delivery [9], yet their brain needs further development before becoming capable of independent survival; (ii) accordingly, the death of a mother reduces the chance of survival of its unweaned dependent infants [10]; (iii) because of the high risk of late-life pregnancy, terminating reproduction and investing in the dependent offspring or grand-offspring may be a better strategy [8,11]. In addition to investing in unweaned dependent descendants, the investment in adult offspring may also be a contributing factor: adult male resident killer whales have higher survival rate [12], and adult female humans have higher fertility rate [10], if their mother is present. In this work, we examined how different investment strategies—including investing in unweaned offspring, unweaned grand-offspring and adult offspring—affect the strength of purifying selection on survival and reproduction in different life stages of female individuals in a population.

Recently, there have been theoretical and *in silico* studies that investigated the link between mother-care/grandmother-care and PPRLS [13–17] (see [18] for review). Some of these models are criticized for oversimplification [18], such as making the assumption of a non-evolvable menopause age [15]. Furthermore, mutation implemented in these models is neutral, making them inadequate to study the decay of purifying selection caused by ageing. Phenotype evolution in these models thus may be partly driven by neutral mutational drift, instead of solely by adaptation. This is contrary to our understanding that mutation is prone to be deleterious [19], and also each intermediate step of the evolutionary pathway of a complex phenotype must be, at least in some biological systems, adaptive [20]. This neutral nature of mutation may have also led to behaviours like 'cost free evolution' that drives the agents 'toward greater and greater longevity' in the models [16]. As a consequence, these models may rely on *a priori* constraint functions on evolvable phenotypes, e.g. a trade-off function between fertility and survival [15,16], as a remedy to the cost-free evolution. Constraint functions like this may, however, be unnecessary if the model can properly account for the deleterious nature of mutation and the decaying strength of purifying selection with increasing age.

Here, we present an agent-based modelling framework adapted from Šajina & Valenzano [21]. It can account for the decaying purifying selection due to ageing on survival and reproduction rates. This modelling framework does not employ any *a priori* trade-off functions to constrain the survival or reproduction rates, instead these rates are evolvable and encoded in the genome of each agent. Mutation, which is prone to be deleterious, drives the value of these rates towards zero and prevents an agent from evolving towards unbounded somatic or reproductive lifespan. Rate parameters critical to fitness are, nevertheless, maintained at high value by strong purifying selection. We applied various strategies of investment into descendants in the model simulations. We showed that, if we properly account for the effects of ageing, PPRLS emerges under some of the investment strategies.

# 2. Material and methods

Our model considers a population of agents that evolves through steps of time; each agent is represented by a genome that encodes its survival and reproduction rates at different life stages. The simulation of a model population continues until the statistical properties of the survival and reproduction rates within the population stabilize; and by performing model simulations under different strategies of intergenerational investment, we investigate how they affect the survival and reproduction in different life stages, and examine under what strategies does it incur the relaxation of selection only on reproduction but not on survival in mid-life. Here, an agent can be male or female. The integer index $i$, $i \in 0, \ldots, 101$, labels the life stage of an agent, and increases by 1 in each time step. Agents at stage $i = 101$ always die. Reproduction starts at $i = 8$. $s_i$ ($i \in 0, \ldots, 100$) and $r_i$ ($i \in 8, \ldots, 100$) of an agent denote its intrinsic survival and reproduction rate in stage $i$, respectively. All $s_i$ and $r_i$ are evolvable, except for $s_i$, $i \in 0, 1$, because these two stages represent unweaned dependent infants that rely upon external provision. This model configuration is critical to the establishment of investment into future generation, and is also consistent with the observation that the presence of mother or grandmother can substantially enhance the survival of human infants in their initial years [17,22,23]. We set the intrinsic survival rate of unweaned infants $s_i = 0.9$, $i \in 0, 1$; using a lower value, e.g. 0.7, may incur rapid extinction. The evolvable intrinsic survival and reproductive rates are encoded in the genome of each agent. They form 192 parameters, $99 = 101 - 2$ for survival and $93 = 101 - 8$ for reproduction, that govern the agent's behaviour.

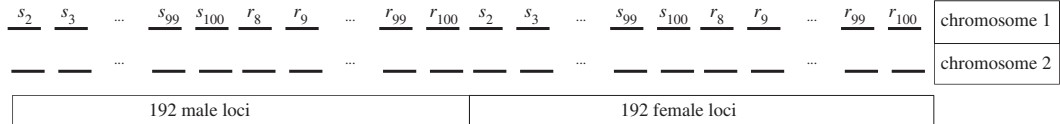

**Figure 1.** Illustration of the genome structure of an agent. A genome has two chromosomes. Each chromosome has a set of 192 male loci and another set of 192 females loci. The 192 loci within a set map to 99 survival rates, $s_i$, life stages $i = 2, \ldots, 100$, and 93 reproduction rates, $r_i$ $i = 8, \ldots, 100$.

The genome of an agent is diploid and has two chromosomes. A chromosome is represented by a sequence of loci that correspond to the evolvable parameters. There is a set of parameters for male, another set for female, and so a chromosome has $384 = 192 \times 2$ loci (figure 1). The parameters for male have no effect if the agent is female, and vice versa. The intrinsic $s_i$ (and $r_i$) of an agent is the average value of the corresponding locus on the two chromosomes.

A simulation starts with 1000 agents that have uniform genome. Each agent has its gender randomly assigned, and its life stage randomly assigned from the range $0 \leq i \leq 50$. Initial parameters encompass $s_i = 0.97$ for $i \geq 2$, $r_i = 0.4$ for $8 \leq i \leq 16$, and $r_i = 0.1$ for $i \geq 17$. Smaller initial values of $s_i$ and $r_i$ sometimes lead to rapid extinction. Moreover, the equilibrium values of $s_i$ and $r_i$ are determined by selection, but not by their initial values. For example, while we used $r_{16} = 0.4$ and $r_{17} = 0.1$ at the outset of each simulation, the average $r_i$ at the end does not show a kink at $i = 16, 17$ in any conditions (figure 3).

In each step, 1000 units of resource are replenished, and each agent consumes one unit. Surplus resource, if there is any, will not be transferred to the next step. Let $N_t$ be the number of agents at step $t$. When $N_t \leq 1000$, an agent at stage $i$ either survives to the next step with probability $s_i$, or dies with probability $1 - s_i$. A female at life stage $i \geq 8$ gives birth to one child with probability $r_i$. To give birth, it pairs up with a random mature male in the population, weighted by the reproduction rate of the male individuals. The gender of the newborn agent is randomly assigned. It receives one arbitrary chromosome from each parent, and thereafter undergoes crossover and mutation. When $N_t > 1000$, famine occurs and the chance to survive and to reproduce is compromised. For simplicity, we set the actual reproduction rate of all agents to 0. The intrinsic death rate of an agent, $1 - s_i$, is magnified by an exponential factor $3^{T_f}$, where $T_f$ is the number of consecutive stages that famine has lasted [21]. The actual survival rate, thus, is $\max(0, 1 - 3^{T_f}(1 - s_i))$.

The size of the simulated populations in this study is maintained to be around 1000, which is much smaller than the real population size of many animal species. Thus, the force of selection acting on the simulated population is weaker than in reality, and the stochastic noise is also stronger. But on the bright side, using such a small population size in a simulation study does not necessarily invalidate the conclusion: if a certain investment strategy incurs PPRLS in our simulation, the same strategy with a lower interaction strength may still be efficacious in a larger population.

The possible values at each locus are discrete. They encompass $0, 0.20, \ldots, 0.60, 0.80, 0.82, \ldots, 0.88, 0.90, 0.91, \ldots, 0.98, 0.99, 0.999$ for $s_i$, and $0.0, 0.1, \ldots, 0.9, 1.0$ for $r_i$. The values of $s_i$ are spaced unevenly, so that a locus drops from a high value to 0 quickly when purifying selection is absent. This configuration saves computational resource and is also realistic. Based on this assumption, the change in fitness in a mutational step gets more negative when it is farther away from the maximum; in other words, the landscape of $s_i$ is roughly convex, which is a common assumption in fitness landscape modelling; using brightness as a proxy indicator of fitness, the fitness landscape of green fluorescent protein, at least in the first few mutational steps from the maximum (wild-type allele), is also convex (see Extended Data fig. 2 of [24]). During crossover, the two chromosomes swap their segments. The chance for a position between two consecutive loci to be a crossover break-point is 10%. After crossover, each locus is mutated at a rate 0.025. We define the ratio between beneficial and deleterious mutation, $\eta$, to be 0.5. A locus chosen to mutate is 50% chance to decrease by one level, $50\% \times \eta = 25\%$ chance to increase by one level, and $50\% \times (1 - \eta) = 25\%$ chance to have no change. This deleterious nature of mutation is consistent with experimental observations [24]. It makes a locus lacking purifying selection to have close-to-zero value.

We considered several strategies of investment in descendants (see table 1 for details). In short, 'NULL' denotes the model without any investment interactions. Mother-care 'M' (grandmother-care 'GM') allows the reduction of death rate of an unweaned dependent infant by 10-fold if its mother (maternal grandmother) is alive. We also imposed two types of long-term cares 'LT' on offspring. An

**Table 1.** List of strategies of investment in offspring and grand-offspring.

| symbol | name | description | original value | new value |
|---|---|---|---|---|
| NULL | basic null model | — | — | — |
| M | mother-care | the presence of mother reduces the death rate of unweaned dependent infant ($i \in 0, 1$) by 10-fold to raise their survival | $s_i$ | $1 - \frac{1-s_i}{10}$ |
| GM | grandmother-care | the presence of maternal grandmother reduces the death rate of unweaned dependent infant ($i \in 0, 1$) by 10-fold to raise their survival | $s_i$ | $1 - \frac{1-s_i}{10}$ |
| LTr | reproduction-enhancing long-term-care | the presence of mother raises the reproduction rate of offspring by twofold ($i \geq 8$) | $r_i$ | $\min(1, 2r_i)$ |
| LTs | survival-enhancing long-term-care | the presence of mother reduces the death rate of offspring ($i \geq 2$) by 10-fold to raise their survival | $s_i$ | $1 - \frac{1-s_i}{10}$ |

independent agent, either weaned infant or adult, has (i) a higher reproduction rate 'LTr', or (ii) a higher survival rate 'LTs', if its mother is alive. These conditions can be combined, e.g. with condition M+GM, the actual death rate of an unweaned dependent infant is reduced by 10-fold if its mother is alive, and by another 10-fold if its maternal-grandmother is alive. The use of 10-fold decrease in unweaned dependent infants' death rate for mother-care is supported by an analysis on Gambian populations, which is found to be 13.4-fold decrease for infants at 0–1 years, and 11.7-fold decrease at 1–2 years [17]. However, the same analysis also reports that grandmother-care incurs a 2.0-fold decrease for infants at 1–2 years. Hence, in the simulation, we used a grandmother-care with an intensity stronger than in reality; this is justified by the need to compensate for the weaker selection force and the stronger stochastic noise due to the unrealistically small population size.

We calculated the survivorship, $l_i$, and individual-fecundity, $m_i$, of the female individuals in the population to infer their overall reproduction and somatic longevity. The survivorship and fecundity at time $t$ are calculated from the statistics of the population sampled within the period $[t - 10\,000, t]$. Specifically, $l_i$ is the probability for a newborn to survive to stage $i$, and $m_i$ is the average reproductive output—in other words, the chance to give birth—of an individual at stage $i$ [25]. PPRLS can be quantified using post-reproduction time (PrT) and post-reproduction representation (PrR) [26]. The average remaining lifespan at stage $i$, $e_i$, is defined as

$$e_i = \frac{\sum_{k=i}^{\infty} (k-i)(l_k - l_{k+1})}{\sum_{k=i}^{\infty} l_k - l_{k+1}}.$$

Let $B$ and $E$ be the smallest integers that satisfy $\sum_{i=0}^{B} m_i \geq 0.05 \sum_{i=0}^{\infty} m_i$ and $\sum_{i=0}^{E} m_i \geq 0.95 \sum_{i=0}^{\infty} m_j$, respectively. $B$ and $E$ represent the stage of beginning-of-reproduction and end-of-reproduction, respectively; let us also denote $E$ as reproductive longevity. Let us define $\mathrm{PrT} = e_E$, the expected lifespan after the end-of-reproduction [26]. PrT is intuitive but vulnerable to statistical noise. Croft *et al.* [18] pointed out that a tiny number of exceptionally long-living individuals in a sample could lead to a high PrT, and hence a false-positive indication of PPRLS. Let us also define [26]

$$\mathrm{PrR} = \frac{l_E e_E}{l_B e_B}.$$

PrR is less intuitive but statistically more robust. PrR is $\ll 0.20$ for species without PPRLS, e.g. 0.02 for wild chimpanzee, and higher otherwise, e.g. 0.22 for resident kill whales, and 0.32–0.71 for different human samples (e.g. [18]). We used 0.20 as the cut-off PrR for PPRLS in this study.

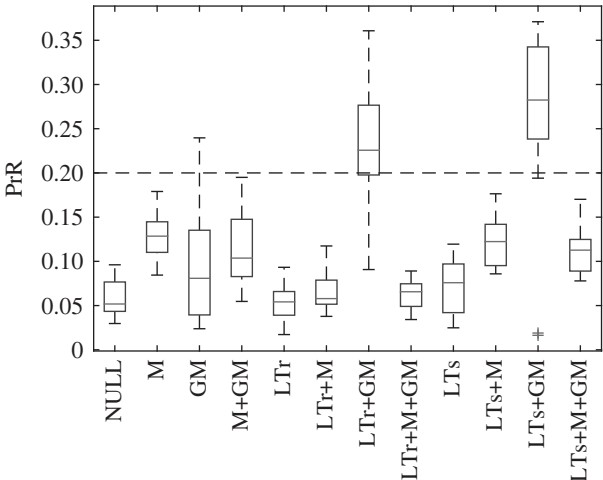

**Figure 2.** Distribution of PrR of conditions NULL, M, GM, M+GM, LTr, LTr+M, LTr+GM, LTr+M+GM, LTs, LTs+M, LTs+GM and LTs+M+GM; their PrR are $0.06 \pm 0.02$, $0.13 \pm 0.02$, $0.10 \pm 0.07$, $0.11 \pm 0.04$, $0.05 \pm 0.02$, $0.07 \pm 0.02$, $0.23 \pm 0.06$, $0.06 \pm 0.02$, $0.07 \pm 0.03$, $0.12 \pm 0.03$, $0.28 \pm 0.09$ and $0.11 \pm 0.02$, respectively. Broken line corresponds to PrR=0.20—the cut-off value for PPRLS in this study. Only LTr+GM and LTs+GM have a PrR distribution with a mean unambiguously beyond 0.20.

## 3. Results and discussion

We simulated different combinations of investment strategies, encompassing the conditions NULL, M, GM, M+GM, LTr, LTr+M, LTr+GM, LTr+M+GM, LTs, LTs+M, LTs+GM, LTs+M+GM (see table 1 for definition of symbols). Every condition is simulated five times, each lasted for 200 000 steps. The relaxation process of every simulation ends within the first 100 000 time steps (electronic supplementary material, figure S1); we calculated the survivorship and fecundity of the female individuals every 20 000 steps, starting from the 100 000th step, to infer PrR and other properties of the simulated population. There are two conditions with PrR unambiguously surpassing 0.20: LTr+GM and LTs+GM (figure 2).

As predicted by the standard evolutionary theory of ageing, strong purifying selection at early ages indeed drives $s_i$ close to one (figure 3). High infant death, however, leaves a sharp kink on survivorship at early stages in some conditions like NULL (figure 4); such sharp kink is also observed in human and whale populations (e.g. [18]). In NULL, the survivorship drops by 60% within the first two life stages, and this appears incongruent with the fixed 0.1 intrinsic death rate in the first two stages. This is because the population size occasionally increases beyond the resource limit, leading to famine and a higher actual death rate.

### 3.1. Mother-care (grandmother-care) alone does not lead to PPRLS

Mother-care (M) effectively protects dependent infants. It alleviates the sharp decrease of survivorship near $i = 2$ and more newborns can survive to adulthood. This consequently weakens the purifying selection on fertility, resulting in a shorter reproductive and somatic lifespan as compared with NULL (figure 4). Nevertheless, the strong purifying selection on survival and reproduction also relaxes simultaneously in late life stages (figure 3). The PrR of the simulated populations does not exceed 0.20 (figure 2), and so mother-care alone does not lead to PPRLS.

As opposed to the strong protection on unweaned dependent infants incurred by mother-care, grandmother-care (GM) alone can only slightly mitigate the low survivorship of unweaned infants (figure 4). Compared to having a living mother, the chance for an infant to have a living grandmother is much lower. Hence, this renders the provision of a grandmother inefficacious.

### 3.2. Scaling equation between infant survival and adult fertility

Investment in descendants affects infant survival and adult fertility of the females in the population. Let us quantify the overall infant survival by $l_2$, and the overall fertility by the total lifetime fecundity,

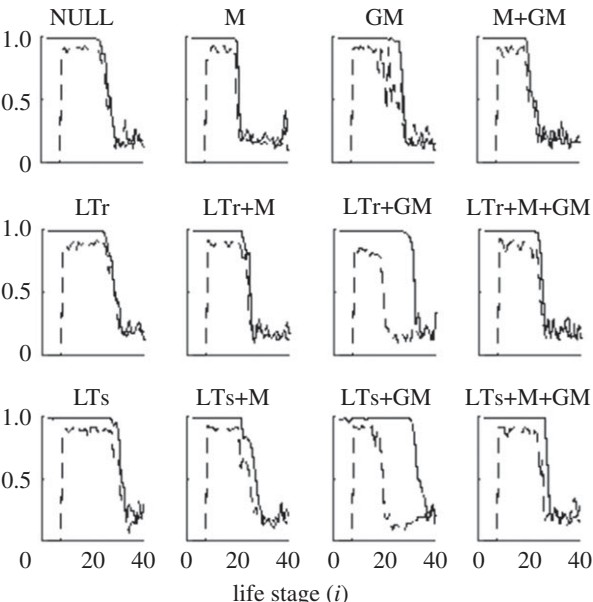

**Figure 3.** Female intrinsic rate of survival $s_i$ (solid curve) and reproduction $r_i$ (broken curve). Each curve is an average over every chromosome at the end of five simulations. The curves of both $s_i$ and $r_i$ decrease sharply and simultaneously, signalling the simultaneous relaxation of purifying selection on survival and reproduction, at late life in all conditions except for LTr+GM and LTs+GM. For the conditions LTr+GM and LTs+GM, the sharp decrease of $s_i$ happens substantially later than the sharp decrease of $r_i$, as the relaxation of purifying selection on survival happens later than the relaxation of selection on reproduction.

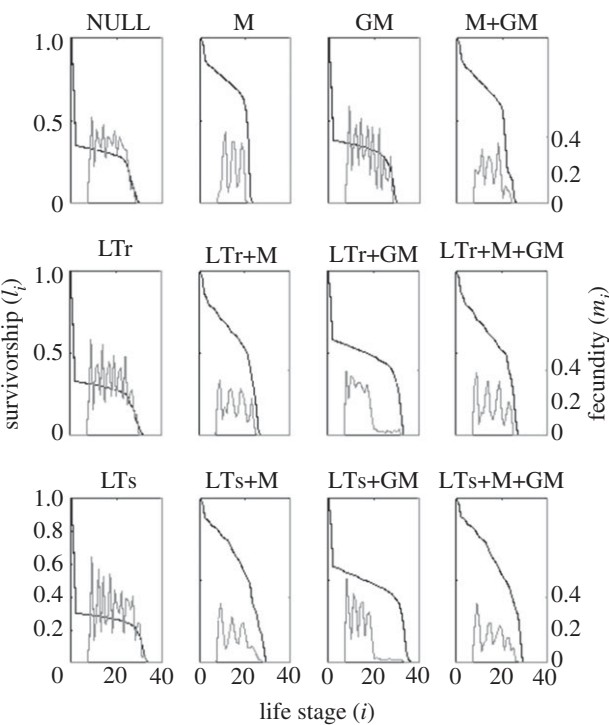

**Figure 4.** Survivorship $l_i$ (black) and individual-fecundity $m_i$ (grey) of female individuals. Each curve is inferred from the statistics sampled at the end of five different simulations. Both $l_i$ and $m_i$ drop to zero almost simultaneously, except for LTr+GM and LTs+GM.

$m_{\text{tot}} = \sum_i m_i$. Infant survival and fertility can be summarized by the phenomenological equation

$$m_{\text{tot}} = \frac{2.33}{l_2}$$

(figure 5*a*), except for the conditions involving mother-care, which slightly deviate from this scaling equation (dot-markers in figure 5). Here, 2.33 is the numerical value of $m_{\text{tot}} \times l_2$, averaged over conditions without mother-care (figure 5*b*). It can be roughly interpreted as the expected number of offspring of an infant

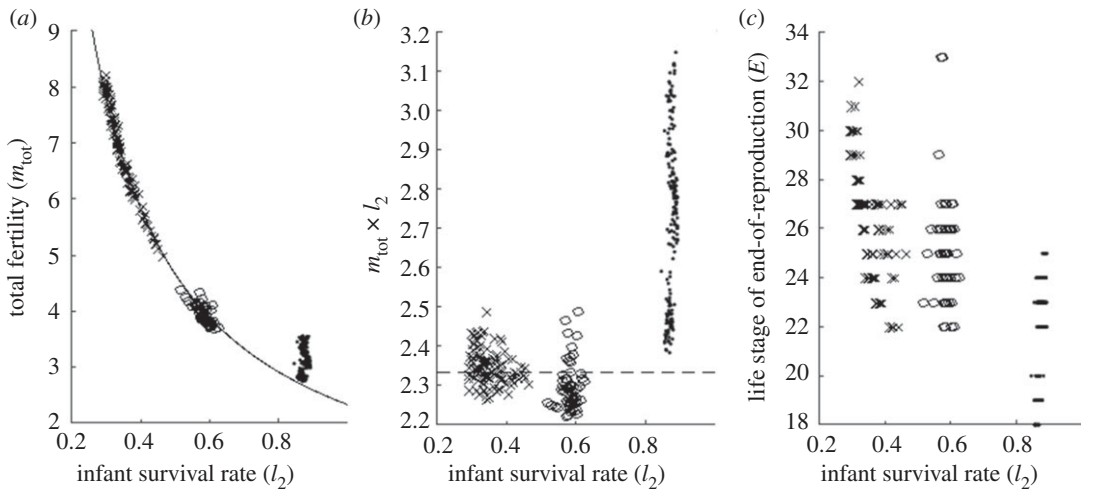

**Figure 5.** (a) Fertility (defined as $m_{tot}$: total lifetime fecundity), (b) the product of fertility and infant survival rate ($l_2$), and (c) the end-of-reproduction life stage (E) plotted against the infant survival rate, defined as $l_2$, i.e. the probability to survive to stage 2. Circle markers are data from conditions that have PPRLS, dots are conditions that involve mother-care, and crosses are the other tested conditions. The solid curve in (a) is $y = 2.33/x$. The broken curve in (b) is $y = 2.33$.

female individual at life stage $i = 2$, and is essentially the equilibrium of a 'tug-of-war' process. On the one hand, as agents with higher fertility have higher fitness, natural selection serves as an upward force on fertility. On the other hand, the deleterious prone mutation serves as a downward force on fertility. When the overall fertility of the population is below certain equilibrium, the upward force is in advance. But when the fertility of the population is beyond the equilibrium, it leads to more frequent famine and a higher death toll. This weakens the upward force, allowing the downward force to dominate and reduce the overall fertility. The fertility of the population hence fluctuates around a stable equilibrium. Furthermore, we also observed a strong correlation between fertility and reproductive lifespan, even though their relationship is far from linear (figure 5c; Spearman correlation: $\rho = 0.8474$, $p < 2.2 \times 10^{-16}$).

## 3.3. Quantifying the efficacy of different investment strategies

Let us quantify the efficacy of an investment strategy at stage $i$ by $c_i$, the average number of care-receivers. $c_i$ of strategies M, LTr, LTs and GM are

$$c_i^M = m_{i-2}l_1 + m_{i-1}l_0,$$

$$c_i^{LTr} = \sum_{j=8}^{i-9} m_j l_{i-j-1},$$

$$c_i^{LTs} = \sum_{j=8}^{i-2} m_j l_{i-j-1}$$

and

$$c_i^{GM} = \sum_{j=8}^{i-1} \sum_{k=0}^{i-1} \frac{m_j}{2} l_k m_k (l_1 \delta_{i-j-k-3} + l_0 \delta_{i-j-k-2}).$$

Here, $\delta_x$ is the Kronecker delta: $\delta_x = 1$ if $x = 0$, and $\delta_x = 0$ if $x \neq 0$. The factor 2 in the denominator of $c_i^{GM}$ accounts for the grand-offspring produced by the daughters but not sons. The efficacy of mother-care at stage $i$, $c_i^M$, scales almost linearly with fecundity $m_i$ and goes to zero at late-life stages. By contrast, the efficacy of the long-term-cares and grandmother-care at stage $i$, $c_i^{LTr}$, $c_i^{LTs}$ and $c_i^{GM}$, are cumulative in nature. They are smaller at earlier life stages, increase as one ages, and do not go to zero even when one can no longer reproduce. This cumulative nature allows an individual to be more contributive to its own reproductive success at late-life than at mid-life. While GM, LTr or LTs alone cannot incur purifying selection on survival that is strong enough to make the somatic lifespan outlast the reproduction lifespan, it works when one combines GM with LTr or LTs. Hence, GM+LTr and GM+LTs are PPRLS-positive.

Interestingly, the conditions LTr+M+GM and LTs+M+GM, which are similar to LTr+GM and LTs+GM except that they have additional mother-care, are PPRLS-negative despite mother-care improving

survivorship of unweaned dependent infants (figure 4). As the chance for the mother to be with an unweaned dependent infant is higher than the chance for the grandmother, mother-care is, therefore, more efficacious than grandmother-care; this is supported by the observation in figure 5a–c that conditions with mother-care (black dots) have infant survivorship ($l_2$) significantly higher than other conditions. Thus, if we switch on mother-care M in condition LTr+GM and LTs+GM, the infants consequently switch their reliance of provision from their grandmother to their mother. This weakens the purifying selection on survival at late life stages and terminates PPRLS.

## 3.4. Model parameterization and sensitivity analysis

Our agent-based model includes a few parameters: population size, mutation rate, crossover rate, survival rate of unweaned dependent infants, starting age of independent infanthood, starting age of reproduction, ratio between beneficial and deleterious mutation, and also various interactions between carer and care-recipient. The numerical value of these parameters are mostly chosen based on the criteria to save computational recourse and to maintain a viable population in the simulations.

We first set the population size to be around 1000, which is far lower than the size of ancestral humans but is computationally affordable. Because of this small population size, we used a per-locus mutation rate (0.025) that is much higher than actual human genes to speed up the simulations, but not too high to incur mutational meltdown and rapid extinction. The crossover rate (0.1) is chosen so that a chromosome is divided into crossover segments that contain tens of loci.

We used a non-evolvable survival rate (0.9) for unweaned dependent infants (stage $i = 0, 1$) and then allowed the mother or maternal grandmother of an infant to enhance its survival. The weakened infant survival created a demand for provision and an opportunity for investment, its value is chosen to be as low as possible without causing rapid extinction. This parameter is also coupled with the starting stage of weaned independent infanthood: if we make the independent infanthood start at a later life stage, then we need to raise the non-evolvable survival rate of unweaned dependent infanthood to avoid rapid extinction. The model is not sensitive to the starting stage of sexual maturity ($i = 8$), and doubling its value has no qualitative change to the survivorship and fertility of the NULL condition.

For the interaction strength between a carer and a care-recipient, we used 10-fold decrease of death rate and twofold increase of birth rate. Using a smaller value for these interactions may reduce the PrR of a PPRLS-positive condition, or even terminate the PPRLS.

We performed further simulations to test the sensitivity of PPRLS-positive conditions, LTr+GM and LTs+GM, by weakening the investment interactions. We considered limited provision in long-term-cares, i.e. limiting reproduction-enhancing long-term-care (LTr) to only the female offspring, and limiting survival-enhancing long-term-care (LTs) to only the male offspring. We simulated the modified LTr+GM and LTs+GM five times, each lasting for 200 000 steps. Under these perturbations, PrR changes from $0.23 \pm 0.06$ to $0.20 \pm 0.09$ for LTr+GM, and from $0.28 \pm 0.09$ to $0.23 \pm 0.08$ for LTs+GM. The reduction of number of care-receivers weakens the PrR signal, but PPRLS nonetheless remains.

We simulated the condition GM, and also the PPRLS-positive condition LTs+GM, using different ratios of beneficial to deleterious mutation, $\eta = 0.1, 0.9$ (default 0.5). At $\eta = 0.1$, mutation tends to be very deleterious and incurs rapid extinction. At $\eta = 0.9$, mutation tends to be mildly deleterious. In GM, the strong purifying selection relaxes at life stage $i \sim 50$ for the survival rates $s_i$, but at stage $i \sim 30$ for the reproduction rates $r_i$ (figure 6), which is not observed at $\eta = 0.5$ s. In LTs+GM, the asynchronous relaxation of purifying selection on survival and reproduction remains when switching from $\eta = 0.5$ to $\eta = 0.9$. But despite this differential relaxation of purifying selection on survival and reproduction, PrR remains less than 0.20 for both GM and LTs+GM at $\eta = 0.9$, because the weakly deleterious mutation allows $r_i$ to stay well above 0 even if a locus is not selected for (figure 6).

All in all, most model parameters are chosen to minimize computational cost without incurring rapid extinction. Parameters critical to PPRLS include the strength of the interactions between carer and care-recipient, and also the ratio between beneficial and deleterious mutation, $\eta$. Our simulations showed that these interaction strengths need to be strong enough to induce PPRLS. The setting of $\eta$ is tricky. When mutation is too deleterious (small $\eta$), rapid extinct occurs. But when mutation is not deleterious enough (large $\eta$), it allows the rate parameters $s_i$ and $r_i$ to be considerably above zero when purifying selection is absent, making PrR small (less than 0.20) even when there is an asynchronous relaxation of purifying selection on survival and reproduction. This is undesirable, because we need to fine tune $\eta$ in order to get a high PrR. A possible remedy to this problem is to include a locus in the chromosome that represents an evolvable age of menopause, such that reproduction terminates even if the reproduction rates thereafter are above zero.

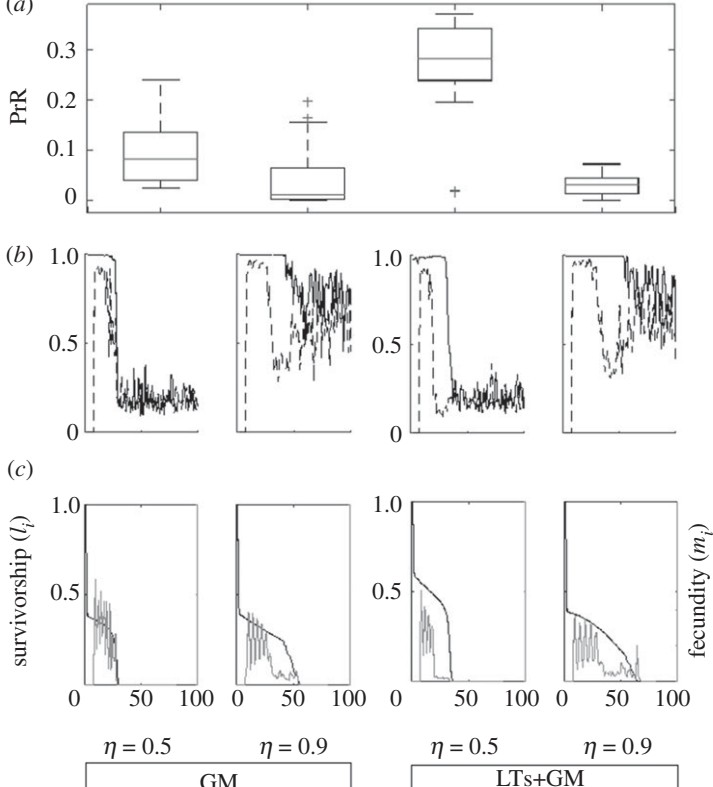

**Figure 6.** (a) Distribution of PrR, (b) intrinsic rate of survival $s_i$ (solid curve) and reproduction $r_i$ (broken curve) averaged over every chromosome at the end of simulations, and (c) survivorship $l_i$ (black) and individual-fecundity $m_i$ (grey) inferred from the statistics sampled at the end of the simulations, for the conditions GM and LTs+GM, with ratio between beneficial and deleterious mutation, $\eta$, equals 0.5 (default) and 0.9. At $\eta = 0.9$, relaxation of purifying selection on survival happens substantially later than relaxation of selection on reproduction on both GM and LTs+GM (b). The PrR of both conditions, however, is <0.20 because the mildly deleterious mutation allows the reproduction rates $r_i$ at late stages to stay substantially above 0 even without purifying selection.

## 4. Conclusion

In sum, we introduced a modelling framework that can account for the deleterious nature of mutation, and quantify age-specific selection on survival and reproduction, and therefore it is suitable for investigating the effects of ageing. We applied this model to test how different strategies of investment in descendants affect relaxation of purifying selection on survival and reproduction. We showed that the caring for unweaned dependent descendants, along with the long-term-caring for independent descendants (infants and adults), are necessary to incur a delay to the relaxation of purifying selection on survival. This prolongs only the somatic lifespan but not the reproductive lifespan, and thus gives rise to PPRLS.

In the sensitivity test, we showed that when mutation is mildly deleterious (large $\eta$), conditions other than LTr+GM or LTs+GM may also lead to asynchronous relaxation of purifying selection on survival and reproduction. A defect in our model design, however, allows the rate parameters to stay well above zero even without purifying selection, making the PrR stay below the detection threshold. Hence, it is worthwhile to test conditions like GM, LTr or LTs again after fixing the defect.

The use of PrR value 0.20 as a cut-off for PPRLS in this study is ambiguous and arbitrary. A thorough survey on the PrR of different animal populations may help to map out the transitional values of PrR, but a theoretical study can further illustrate the subtle details behind the transition. Specifically, we can use the proposed model to find out the PrR values over a broad range of offspring-care intensity, and also over a wide scope of offspring-care strategies. Moreover, the environment of the simulated population is also a variable, e.g. instead of a stable environment with constant supply of resource, we can replace it with one that has an expanding, shrinking, or fluctuating resource supply, and examine whether PPRLS is a consequence of different strategies.

Our proposed modelling framework can further serve as a supplement to the kin selection framework. Kin selection, a school of hypotheses alternative to the infant-care hypotheses, analyses how an individual's social interactions, such as migration and dispersion, or selection conflict between

kins, shape the scheduling of reproduction and menopause, and also the longevity of a species [27–30]. Many of these models rely on the concept of inclusive fitness to quantify the benefit of different social behaviours; here, the use of inclusive fitness is based on the assumption that nature selects for behaviours that can maximize an individual's and also its kins' offspring output [31,32]. In the light of those pioneering studies, it is worthwhile to investigate their findings using our modelling framework, and examine the role of age-specific selection therein.

In the earliest versions of mother-care and grandmother-care hypothesis, it was posited that females switch off reproduction in mid-life while maintaining the somatic lifespan intact [8]. Later, comparative studies on humans and great apes showed that both have similar ages of last-birth. This shifted our view from stop-reproduction-earlier to die-later [33,34]. Based on these developments, our modelling framework further provides a more detailed and quantitative description on how infant survival, somatic lifespan and reproductive lifespan depend on the investment strategies of intergenerational transfer. Our simulation also revealed a scaling law between fertility and infant survival. There are other hypotheses to be explored, e.g. the preference of males for younger females is hypothesized to be a driving force to the cessation of reproduction in mid-life [35]. We are left with many questions: Do other hypotheses work if ageing is properly accounted for? How did our ancestral species adaptively develop to these investment strategies? Our model is an appropriate tool to answer these questions, and we leave them to future studies.

Data accessibility. The source code scripts, along with the raw data of the model simulations under different conditions, are compressed into a single file of electronic supplementary material. The usage of these script files and the format of the raw data are explained in the README files therein. The electronic supplementary material is stored in the Dryad Digital Repository and available at https://doi.org/10.5061/dryad.vt4b8gtnd [36].

Competing interests. We declare we have no competing interest.

Funding. This work was supported by SFB 680 and SFB 1310 of German Research Foundation (DFG), Volkswagen Funding (VolkswagenStiftung) initiative 'A unified model of recombination in life', and Heinrich Heine University Düsseldorf.

Acknowledgements. We thank Sergei Maslov, Dario Valenzano and Chik Him Wong for helpful comments.

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
