## [Reviewer comments · Royal Society Open Science]

Review History

RSOS-191972.R0 (Original submission)

Review form: Reviewer 1

Is the manuscript scientifically sound in its present form?

Yes

Are the interpretations and conclusions justified by the results?

No

Is the language acceptable?

Yes

Do you have any ethical concerns with this paper?

No

Have you any concerns about statistical analyses in this paper?

Yes

Recommendation?

Accept with minor revision (please list in comments)

Comments to the Author(s)

The ms examines the evolution of age-specific survival and reproduction in different scenarios of maternal and grandmaternal care on offspring reproduction and survival using agent-based simulations. The results suggest that only grandmaternal care with long-term caring of offspring can select for marked post-reproductive survival (measured as PrR).

Introduction

The information given in the first paragraph sounds a bit vague. For example, evolution is “only” a process of differing gene frequencies between the subsequent generations and does not “select” for anything. The selecting force is natural selection acting on individuals or gene(s). Likewise, the reason for weaker selection at older ages is the expected lower reproductive output and survival prospects of old individuals compared to younger ones, not their absolute numbers in the population (which is also usually true in natural populations, except in western modern human populations with a skewed age pyramid).

The hypothesis based on reproductive conflicts between the maternal generations (Cant & Johnstone 2008 Reproductive conflict and the separation of reproductive generations in humans. PNAS 105, 5332-5333) seems to be missing from the study. Any justification for that or the implications it might have to the performance of the current simulation?

I am not familiar with agent-based modelling so I suggest someone expert in this approach takes a good look at the details of the model. Would it make sense to explain this simulation framework by couple of (common sense) lines before going into the details of model specification? Based on my reading it seems that the model assumes that reproduction starts at the age-class (i) of 8 time steps and every agent is dead by the “age” of 101 time steps. But it also seems that agents can reproduce until the age of 100 time steps? And the contribution of a mother or a grandmother is a 10-fold increase in survival rate? How realistic are these parameters if the goal is understand the evolution of human life histories. And how should one understand the statements that some smaller initial values lead to rapid extinction and were thus not considered? That your initial population size (which seems to act like some sort of a carrying capacity for the populations) was just too small? And so what if such extinctions sometimes happen, is that not what has and will happened to mutations/lineages also in the real life? Moreover, I have no idea what lines 104-105 actually imply, please clarify. All these questions may just reflect my incompetence with respect to the modeling approach rather than real problems in the simulations.

In line 120, it is said that: “The values of s_i are spaced unevenly, so that a locus drops from a high value to 0 quickly when purifying selection is absent to save computational resource.” Are these models really so heavy for modern computers (or superclusters etc.) that one is forced to make unrealistic assumptions to complete the runs?

How robust are the conclusion for the PrR cut-off of 0.2 used in the simulations? I mean did decreasing or increasing this threshold by e.g. 0.05 have any influence on the results while keeping in mind that species with $\text{PrR} < 0.2$ are considered to have no post-reproductive stage while species with $\text{PrR} = 0.22$ might have one (killer whales)? Some answers to this may be provided in Fig. 2 but since it does not provide explanation for the error bars of PrR (just “distribution” is mentioned) of different setups, it is hard to judge.

How these results deal with changing population dynamics? I think early reproduction is particularly favored in growing populations, so one might expect such conditions to not greatly favor long lifespan and grandparenting (and late menopause). Whereas in more stable population sizes, reproduction at older ages also makes a strong contribution to relative fitness and grandparenting could be more favored by selection in such conditions.

In the end (i.e. the conclusion paragraph), it is unclear to the reader how the new modeling results correspond to the previous ones? For example, are there some contrasting findings or conclusions, did these attempts use the same simulation framework etc.

Review form: Reviewer 2

Is the manuscript scientifically sound in its present form?

Yes

Are the interpretations and conclusions justified by the results?

Yes

Is the language acceptable?

Yes

Do you have any ethical concerns with this paper?

No

Have you any concerns about statistical analyses in this paper?

No

Recommendation?

Accept as is

Comments to the Author(s)

Dear Editor,

The manuscript "On age-specific selection and extensive lifespan beyond menopause" presents an elegant and interesting agent-based model of a population undergoing the opposing forces of accumulating deleterious mutation and purifying selection. The author models how these forces affect a species expected survival and reproductive lifespan.

The author considers various scenarios of mother care and grandmother care and combinations of these strategies influence the evolution of prolonged post-reproductive lifespans.

The work is innovative, clearly explained, and form a strong basis for future investigation, so I think it is worthy of publication.

Decision letter (RSOS-191972.R0)

07-Feb-2020

Dear Dr Pang

On behalf of the Editors, I am pleased to inform you that your Manuscript RSOS-191972 entitled "On age-specific selection and extensive lifespan beyond menopause" has been accepted for publication in Royal Society Open Science subject to minor revision in accordance with the referee suggestions. Please find the referees' comments at the end of this email.

The reviewers and handling editors have recommended publication, but also suggest some minor

revisions to your manuscript. Therefore, I invite you to respond to the comments and revise your manuscript.

- Ethics statement

- Data accessibility

<http://datadryad.org/submit?journalID=RSOS&manu=RSOS-191972>

- Competing interests

- Authors' contributions

- Acknowledgements

- Funding statement

Please ensure you have prepared your revision in accordance with the guidance at <https://royalsociety.org/journals/authors/author-guidelines/> -- please note that we cannot publish your manuscript without the end statements. We have included a screenshot example of

the end statements for reference. If you feel that a given heading is not relevant to your paper, please nevertheless include the heading and explicitly state that it is not relevant to your work.

Because the schedule for publication is very tight, it is a condition of publication that you submit the revised version of your manuscript before 16-Feb-2020. Please note that the revision deadline will expire at 00.00am on this date. If you do not think you will be able to meet this date please let me know immediately.

If your manuscript is newly submitted and subsequently accepted for publication, you will be asked to pay the article processing charge, unless you request a waiver and this is approved by Royal Society Publishing. You can find out more about the charges at <https://royalsocietypublishing.org/rsos/charges>. Should you have any queries, please contact openscience@royalsociety.org.

on behalf of Kevin Padian (Subject Editor)
openscience@royalsociety.org

Associate Editor Comments to Author (Mr Andrew Dunn):

Associate Editor: 1

Comments to the Author:

Thank you for submitting your research to Royal Society Open Science. As you can see, the reviewers are very supportive of your manuscript, but provide some additional comments and queries that we ask that you please address in a revised version. Please ensure to provide a point-by-point response to their comments, as well as a tracked changes version of your manuscript to highlight the edits made. We look forward to receiving your revised manuscript.

Reviewer comments to Author:

Reviewer: 1

Comments to the Author(s)

The ms examines the evolution of age-specific survival and reproduction in different scenarios of maternal and grandmaternal care on offspring reproduction and survival using agent-based simulations. The results suggest that only grandmaternal care with long-term caring of offspring can select for marked post-reproductive survival (measured as PrR).

Introduction

The information given in the first paragraph sounds a bit vague. For example, evolution is “only” a process of differing gene frequencies between the subsequent generations and does not “select” for anything. The selecting force is natural selection acting on individuals or gene(s). Likewise, the reason for weaker selection at older ages is the expected lower reproductive output and survival prospects of old individuals compared to younger ones, not their absolute numbers in the population (which is also usually true in natural populations, except in western modern human populations with a skewed age pyramid).

The hypothesis based on reproductive conflicts between the maternal generations (Cant & Johnstone 2008 Reproductive conflict and the separation of reproductive generations in humans. PNAS 105, 5332-5333) seems to be missing from the study. Any justification for that or the implications it might have to the performance of the current simulation?

I am not familiar with agent-based modelling so I suggest someone expert in this approach takes a good look at the details of the model. Would it make sense to explain this simulation framework by couple of (common sense) lines before going into the details of model specification? Based on my reading it seems that the model assumes that reproduction starts at the age-class (i) of 8 time steps and every agent is dead by the “age” of 101 time steps. But it also seems that agents can reproduce until the age of 100 time steps? And the contribution of a mother or a grandmother is a 10-fold increase in survival rate? How realistic are these parameters if the goal is understand the evolution of human life histories. And how should one understand the statements that some smaller initial values lead to rapid extinction and were thus not considered? That your initial population size (which seems to act like some sort of a carrying capacity for the populations) was just too small? And so what if such extinctions sometimes happen, is that not what has and will happened to mutations/lineages also in the real life? Moreover, I have no idea what lines 104-105 actually imply, please clarify. All these questions may just reflect my incompetence with respect to the modeling approach rather than real problems in the simulations.

In line 120, it is said that: “The values of s_i are spaced unevenly, so that a locus drops from a high value to 0 quickly when purifying selection is absent to save computational resource.” Are these models really so heavy for modern computers (or superclusters etc.) that one is forced to make unrealistic assumptions to complete the runs?

How robust are the conclusion for the PrR cut-off of 0.2 used in the simulations? I mean did decreasing or increasing this threshold by e.g. 0.05 have any influence on the results while keeping in mind that species with PrR <0.2 are considered to have no post-reproductive stage while species with PrR = 0.22 might have one (killer whales)? Some answers to this may be provided in Fig. 2 but since it does not provide explanation for the error bars of PrR (just “distribution” is mentioned) of different setups, it is hard to judge.

How these results deal with changing population dynamics? I think early reproduction is particularly favored in growing populations, so one might expect such conditions to not greatly favor long lifespan and grandparenting (and late menopause). Whereas in more stable population sizes, reproduction at older ages also makes a strong contribution to relative fitness and grandparenting could be more favored by selection in such conditions.

In the end (i.e. the conclusion paragraph), it is unclear to the reader how the new modeling results correspond to the previous ones? For example, are there some contrasting findings or conclusions, did these attempts use the same simulation framework etc.

Reviewer: 2

Comments to the Author(s)

Dear Editor,

The manuscript “On age-specific selection and extensive lifespan beyond menopause” presents an elegant and interesting agent-based model of a population undergoing the opposing forces of accumulating deleterious mutation and purifying selection. The author models how these forces affect a species expected survival and reproductive lifespan.

The author considers various scenarios of mother care and grandmother care and combinations of these strategies influence the evolution of prolonged post-reproductive lifespans.

The work is innovative, clearly explained, and form a strong basis for future investigation, so I think it is worthy of publication.

Author's Response to Decision Letter for (RSOS-191972.R0)

See Appendix A.

Decision letter (RSOS-191972.R1)

02-Apr-2020

Dear Dr Pang,

It is a pleasure to accept your manuscript entitled "On age-specific selection and extensive lifespan beyond menopause" in its current form for publication in Royal Society Open Science.

on behalf of Prof Kevin Padian (Subject Editor)
openscience@royalsociety.org

Appendix A

Tin Yau, Pang, PhD
Bioinformatics
Heinrich Heine University Düsseldorf
Universitätsstraße 1
40221 Düsseldorf, Germany
Tel: +49-211-81-11651
Fax: +49-211-81-15767
Email: pang@hhu.de

24th, March, 2020

Subject: Submission of revised paper RSOS-191972

Dear Editor,

Thank you very much for the email dated 7th, February, 2020, which encloses the comments of the reviewers. I have carefully revised the manuscript. Below is a point-by-point reply that I made. The comments and questions raised by the reviewers are underlined.

Yours faithfully,
Tin Yau Pang

Reviewer comments to Author:

Reviewer: 1

Comments to the Author(s)

The ms examines the evolution of age-specific survival and reproduction in different scenarios of maternal and grandmaternal care on offspring reproduction and survival using agent-based simulations. The results suggest that only grandmaternal care with long-term caring of offspring can select for marked post-reproductive survival (measured as PrR).

Introduction

The information given in the first paragraph sounds a bit vague. For example, evolution is “only” a process of differing gene frequencies between the subsequent generations and does not “select” for anything. The selecting force is natural selection acting on individuals or gene(s). Likewise, the reason for weaker selection at older ages is the expected lower reproductive output and survival prospects of old individuals compared to younger ones, not their absolute numbers in the population (which is also usually true in natural populations, except in western modern human populations with a skewed age pyramid).

I thank the reviewer for pointing out the ambiguities in the introductory paragraph. I followed this advice and did the modification.

The hypothesis based on reproductive conflicts between the maternal generations (Cant & Johnstone 2008 Reproductive conflict and the separation of reproductive generations in humans. PNAS 105, 5332-5333) seems to be missing from the study. Any justification for that or the implications it might have to the performance of the current simulation?

I would like to thank the reviewer for pointing out my negligence to mention reproductive conflict, which is another important hypothesis. Reproductive conflict between related kins is a major theme of kin-selection, an alternative school of hypotheses that can also explain prolonged PRLS. The framework of kin-selection models is based on inclusive fitness, where the objective of an individual is to maximize its own and also its relatives' offspring output, whereas our modelling framework concerns the presence and absence of age-specific selection to resist deleterious mutations. I have modified the paragraphs in the Discussion section to relate the current study to kin-selection.

I am not familiar with agent-based modelling so I suggest someone expert in this approach takes a good look at the details of the model. Would it make sense to explain this simulation framework by couple of (common sense) lines before going into the details of model specification? Based on my reading it seems that the model assumes that reproduction starts at the age-class (i) of 8 time steps and every agent is dead by the "age" of 101 time steps. But it also seems that agents can reproduce until the age of 100 time steps? And the contribution of a mother or a grandmother is a 10-fold increase in survival rate? How realistic are these parameters if the goal is understand the evolution of human life histories. And how should one understand the statements that some smaller initial values lead to rapid extinction and were thus not considered? That your initial population size (which seems to act like some sort of a carrying capacity for the populations) was just too small? And so what if such extinctions sometimes happen, is that not what has and will happened to mutations/lineages also in the real life? Moreover, I have no idea what lines 104-105 actually imply, please clarify. All these questions may just reflect my incompetence with respect to the modeling approach rather than real problems in the simulations.

I would like to thank the reviewer for raising these questions, as it may confuse readers without first hand experiences on computational simulation. When first preparing this model, I interpreted a time step in the simulation as one year. However, one can also interpret a time step as one month or one decade, and what we need is to pay attention to how the interpretation affects the other settings in the model. Here is the caveat: the model subtly assumes that a female individual can reproduce no more than one child in each time step. Hence, ignoring the case of multiple offspring produced by the same pregnancy, a time step should be considered as the minimum time span between two birth events, which is one or two years.

Moreover, life stage i in the model is allowed to be as high as $i=101$, but this value serves rather like an upper bound than the actual end-of-life stage. The value 101 is arbitrary, as it

can be any number >35 —the life stage where the survivorship and fecundity decay to 0 for all conditions considered (see Figure 4).

I tried to set the parameters as realistic as possible, and put in as few assumptions as possible into the model. However, due to the high computational cost, we can only use an unrealistically small population size. The use of a small population size weakens selection and also leads to a strong stochastic noise, which may overshadow the effect of inter-generational investment. But on the bright side, if simulation using such a small population can show the efficacy of certain infant-caring interactions like grandmother-care and long-term mothercare, then in a larger population, which has a lower stochastic noise level, the same interactions will also be as efficacious even with a lower interaction strength. Therefore, the use of a small population size in the simulations does not invalidate the findings.

Depending upon the conditions, the presence of mother or grandmother may reduce the ad-hoc death rate by 10-fold in the model. Empirically, a regression analysis on Gambian village populations estimated that mothercare can reduce the mortality by a factor of 4 to 44 for infants with age 0 to 1, and by a factor of 3 to 54 for those with age 1 to 2; grandmother-care can reduce the mortality by a factor 1.2 to 3.5 for infants with age 1 to 2 (see Shanley et al., Proceedings B, 2007). However, Shanley et al. also pointed out that, based on theoretical analysis, caring for unweaned infants alone is not sufficient to incur menopause, and one needs to resort to other investment interactions, such as mother / grandmother caring for older offspring; but these interactions cannot be statistically significantly inferred from the data. Further, as we are using a very small population size, we need a much stronger interaction strength in order to overcome the strong stochastic noise faced by the weak selection force. As explained in the previous paragraph, the use of strong intensity for infant-care does not invalidate the finding here.

When I performed the model simulation, I expected that the s_i and r_i in the genomes of the population would gradually converge to certain distribution, and their statistical properties would stabilize. The goal of model simulation is to examine under what settings—e.g., investment strategies, infant death rate—would incur prolonged PRLS. Due to the stochastic nature of simulation, a population can sometimes go extinct. But in case of extinction using a low infant survival rate, the simulation goes extinct five times out of five trials, and hence this the model setting is inherently unstable and therefore unrealistic. This is different from the extinctions of isolated groups within a species, where the species as a whole survives and continues. Therefore, I maintained the infant survival rate to be high enough for a sustainable population.

Line 104-105 says, “For example, while we used $r_{16} = 0.4$ and $r_{17} = 0.1$ at the outset of each simulation, the average r_i at the end of the simulation does not show a kink at $i = 16, 17$ in any conditions (see Fig. 3)”. Let me rephrase it here. At the start of the simulation, the reproduction rate at $i=16$ is set to be 0.4, and the reproduction rate at $i=17$ is set to be 0.1; there is a sudden drop of reproduction rate at $i=16,17$. As the simulation continues, the statistical properties of the population gradually converge and stop changing. The reproduction rates in the population converge to a certain equilibrium; its trend gets smooth

at $i=16,17$, and the step disappears. Hence, as long as the population does not go extinct, the initial values of r_{16} and r_{17} are irrelevant and do not affect the final results.

I have modified the manuscript to address these questions.

In line 120, it is said that: “The values of s_i are spaced unevenly, so that a locus drops from a high value to 0 quickly when purifying selection is absent to save computational resource.” Are these models really so heavy for modern computers (or superclusters etc.) that one is forced to make unrealistic assumptions to complete the runs?

While this assumption is artificial, it is far from unrealistic. In my model, I assumed that the more mutational steps it is away from the optimal value ($s_i=0.999$), the larger the drop in a single step. In other words, I assumed that the fitness landscape of s_i to be roughly convex, which is a common assumption in theoretical modelling of fitness landscape, and is also consistent with measurements on green fluorescent protein (GFP) (Sarkisyan et al., Nature, 2016): using brightness as a proxy indicator of fitness, it is shown that, at least for the first few mutational steps from fitness maximum, the landscape is convex in nature (Extended Data Figure 2).

I have modified the manuscript to explain the negative epistasis assumption more clearly.

How robust are the conclusion for the PrR cut-off of 0.2 used in the simulations? I mean did decreasing or increasing this threshold by e.g. 0.05 have any influence on the results while keeping in mind that species with $\text{PrR} < 0.2$ are considered to have no post-reproductive stage while species with $\text{PrR} = 0.22$ might have one (killer whales)? Some answers to this may be provided in Fig. 2 but since it does not provide explanation for the error bars of PrR (just “distribution” is mentioned) of different setups, it is hard to judge.

The reviewer raised the concern on how susceptible is our result to the uncertainty of the PrR cutoff for prolonged PRLS. We used $\text{PrR}=0.20$ as the cutoff for prolonged PRLS. The two conditions reported in the manuscript to have prolonged PRLS, LTr+GM and LTs+GM, have PrR values 0.23 ± 0.06 and 0.28 ± 0.09 , respectively; hence raising the cutoff value by 0.05—from 0.20 to 0.25—will definitely abrogate the prolonged-PRLS-positivity of LTr+GM. However, despite the uncertainty on the value of the PrR cutoff, we can still visually classify the PrR distributions in Figure 2 into two groups, such that LTr+GM and LTs+GM stand out from all other conditions. Meanwhile, I will put down the mean and standard deviation of different conditions in the figure caption for better information.

The more important point here is that, we did not have a clear picture on how to decide the PrR cutoff, and thus I used an arbitrary value 0.20 in the manuscript. A better definition can be determined if we have a more thorough understanding on this transition. For example, we can study, using my model, how varying the strength of infant-care affects the distribution of PrR. I added this point at the end of the manuscript.

How these results deal with changing population dynamics? I think early reproduction is particularly favored in growing populations, so one might expect such conditions to not

greatly favor long lifespan and grandparenting (and late menopause). Whereas in more stable population sizes, reproduction at older ages also makes a strong contribution to relative fitness and grandparenting could be more favored by selection in such conditions.

I would like to thank the reviewer for possible application of the proposed model. I agree with the reviewer's argument that, in an expanding population, one might find grandparenting to be less helpful and may not be selected for. I have not performed any test on this idea, but this is certainly a good topic that can be addressed using my model.

I modified the manuscript to include this suggestion on the future application of this model.

In the end (i.e. the conclusion paragraph), it is unclear to the reader how the new modeling results correspond to the previous ones? For example, are there some contrasting findings or conclusions, did these attempts use the same simulation framework etc.

The main selling point of my model is that prolonged PRLS emerges in conditions that involve grandmother-care and long-term offspring-care. A few other models, such as those by Kim et al., require additional trade-off functions that forces reproduction or survival to decay with age, or otherwise the agents in the simulation will evolve to unrealistically long somatic and reproductive lifespan; however, after accounting for the deleterious nature of mutation, my model does not require this kind of unnatural trade-off functions.

In the new version of the manuscript, I added the ideas that the reviewer inspired me, including the use of the model to 1) map out how varying infant-care interaction strength affect the distribution of PrR, i.e., the transition to prolonged PRLS, and 2) examine the role of age-specific selection on other possible investment interactions, including those from the kin-selection models.

Reviewer: 2

Comments to the Author(s)

Dear Editor,

The manuscript "On age-specific selection and extensive lifespan beyond menopause" presents an elegant and interesting agent-based model of a population undergoing the opposing forces of accumulating deleterious mutation and purifying selection. The author models how these forces affect a species expected survival and reproductive lifespan.

The author considers various scenarios of mother care and grandmother care and combinations of these strategies influence the evolution of prolonged post-reproductive lifespans.

The work is innovative, clearly explained, and form a strong basis for future investigation, so I think it is worthy of publication.

Reply:

I would like to thank the reviewer for going through my manuscript.

The reviewer did not raise any question.